

# Modafinil decreases anxiety-like behaviour in zebrafish

Adrian Johnson[1] and Trevor James Hamilton[1,2]

[1] Department of Psychology, MacEwan University, Edmonton, Alberta, Canada
[2] Neuroscience and Mental Health Institute, Universtiy of Alberta, Edmonton, Canada

## ABSTRACT

Modafinil (2-((diphenylmethyl)sulfinyl)acetamide), a selective dopamine and norepinephrine transporter inhibitor, is most commonly prescribed for narcolepsy but has gained recent interest for treating a variety of disorders. Zebrafish (*Danio rerio)* are becoming a model of choice for pharmacological and behavioural research. To investigate the behavioural effects of modafinil on anxiety, we administered doses of 0, 2, 20, and 200 mg/L for 30 minutes then tested zebrafish in the novel approach test. In this test, the fish was placed into a circular arena with a novel object in the center and motion-tracking software was used to quantify the time the fish spent in the outer area of the arena (thigmotaxis zone), middle third of the arena (transition zone) and center of the arena, as well as total distance traveled, immobility and meandering. Modafinil caused a decrease in time spent in the thigmotaxis zone and increased time spent in the transition zone across all doses. Modafinil did not significantly alter the time spent in the center zone (near the novel object), the distance moved, meandering, or the duration of time spent immobile. We also validated this test as a measure of anxiety with the administration of ethanol (1%) which decreased time spent in the thigmotaxis zone and increased time spent in the transition zone. These results suggest that modafinil decreases anxiety-like behaviour in zebrafish.

## INTRODUCTION

Modafinil (2-((diphenylmethyl)sulfinyl)acetamide) (brand name Alertec in Canada, Provigil in the United States, and Modavigil in Australia) is a psychostimulant primarily used by narcolepsy patients and shift workers to alleviate sleep related disorders. It is being tested as a potential treatment for major depressive disorder (MDD), cocaine-addiction, and as a cognitive enhancer (*Abolfazli et al., 2011*; *Dean et al., 2011*; *Minzenberg & Carter, 2008*). Modafinil is thought to inhibit dopamine transporters (*Madras et al., 2006*; *Volkow et al., 2009*) as well as and norepinephrine transporters (*Madras et al., 2006*) leading to increased synaptic dopamine and norepinephrine, respectively, but not to a level stimulating abuse of the drug. It is shown to reduce GABA (y-aminobutyric acid) levels, and increase levels of serotonin (5HT), glutamate, orexin, and histamines in the brain (*Minzenberg & Carter, 2008*; *Mereu et al., 2013*). Because of its action on these neurotransmitter systems and low abuse profile, modafinil has the potential for a wide therapeutic benefit.

Corresponding author
Trevor James Hamilton,
trevorjameshamilton@gmail.com

Modafinil has a variety of effects on anxiety in humans and anxiety-like behaviour in animal models. Some studies have shown anxiety generating (anxiogenic) effects in humans, while others show anxiety reducing (anxiolytic) effects. In an emotion and cognition test, *Rasetti et al. (2010)* found that seven days of modafinil (100 mg/day) increased the efficacy of prefrontal cognitive information processing in humans, while reducing the reactivity to fearful and threatening stimuli through the amygdala (controlling anxiety). However, in other studies, repeated doses of modafinil given to sleep apnea patients (200–400 mg/day, four weeks; *Schwartz, Hirshkowitz & Erman, 2003*), and narcoleptic patients (400 mg/day, two weeks; *Broughton et al., 1997*) caused an increase in nervousness and/or anxiety. In another study, *Randall et al. (2003)* found that doses of 100–800 mg in 'healthy young volunteers' had significant anxiogenic effects. Therefore, it is still inconclusive as to what type of anxiety-altering response modafinil might produce in humans.

Often we can move to animal models to find answers to pharmacological questions, however, animal models have also shown differing effects of modafinil. In rhesus macaque monkeys (*Macaca mulatta*), single doses of modafinil have been shown to increase nocturnal activity but did not decrease anxiety responses (*Hermant, Rambert & Duteil, 1991*). *Van Vliet et al. (2006)* found an anxiolytic response in marmoset monkeys (*Callithrix jacchus*) when recording their startle responses to a threat situation after a single oral dose of modafinil (between 50–225 mg/kg). In male swiss albino mice (*Mus musculus*), *Simon, Panissaud & Costentin (1994)* compared modafinil to dexamphetamine (another known stimulant) and measured the relative stimulant properties as well as anxiety levels generated by these substances. Of the three tests used (black and white compartment test, elevated plus maze, and open field task), modafinil did not increase any anxiety behaviours. Interestingly, modafinil dose-dependently increases spontaneous exploration in C57BL/6J mice (*Young, Kooistra & Geyer, 2011*). These studies in animal models also demonstrate an inconsistent effect of modafinil on anxiety and exploratory behaviour.

Zebrafish (*Danio rerio*), have become a popular animal model used in the scientific community due to their fecundity, larval-stage transparency, short gestational period, and ease in handling (*Gerlai, 2010*; *Stewart et al., 2014*). The similarity in zebrafish genetic and behavioural markers to those in humans make this species an ideal model organism for studying pharmacological compounds (*Spence et al., 2008*; *Tierney, 2011*). To date only one study examined the effect of modafinil on zebrafish behaviour and it was performed on larval zebrafish (six days post-fertilization). *Sigurgeirsson et al. (2011)* found that modafinil causes a dose-dependent reduction in sleep. Specifically, they found that modafinil had no effect on the number of transitions between sleep and wakefulness throughout a 24 h monitoring period, but had a decrease in the mean sleep percentages of fish as the dosages increased, especially in the 12 h in which the lights were off at night. Modafinil increased the wakeful bouts of fish whilst maintaining the structure of sleep; however, these researchers did not perform any cognitive or anxiety-related behavioural tests. To date, the effects of modafinil in larval or adult zebrafish remains relativity unexplored.

In this study we used the novel approach test, which is a test of anxiety in zebrafish (*Stewart et al., 2012*) and other fish species (*Ou et al., 2015*). In this test, the fish is placed

into a circular arena with a novel object in the center. Typically, zebrafish are fearful of the novel object (neophobic) and spend time near the wall of the arena (thigmotaxis). With a camera-based motion-tracking software system we quantified the time near the object, near the wall, and in a transition zone in the middle, as well as distance moved, meandering, and immobility, after a 30 min exposure to modafinil (2, 20, and 200 mg/L) or ethanol (1%).

## MATERIALS AND METHODS

The experimental procedures were approved by the MacEwan University Animal Research Ethics Board (AREB; protocol number 05-12-13). These standards are in compliance with the Canadian Council for Animal Care (CCAC).

### Subjects

This study used 138 adult wild-type zebrafish (*Danio rerio)* that were held in a 3-shelf benchtop housing system (Aquatic Habitats) in either 3 or 10 L polypropylene tanks. The fish were obtained from Aquatic Imports (Calgary, AB) and the gender was unidentified. Habitat water was made in the lab by purifying tap water through reverse osmosis (RO), then adding 5 mL of prime (sodium thiosulphate), 25 g of aquarium salt, 10–18 g of NaHCO$_3$, and 100 mL of acetic acid were added for every 60 L of RO water (*Holcombe et al., 2013*). The water in the habitat was continuously recirculated, treated with activated carbon, and was UV irradiated and filtered. The zebrafish were fed once daily on an alternating diet of either flake food (Gamma-micro 300; New Life Products) or shrimp (Omega One Freeze Dried Shrimp Nutri-Treat; Omega Sea Ltd.). Fish were kept on a daily 12-hour light/dark cycle with lights turning on at 8 AM and off at 8 PM. The water conditions in the habitat were monitored daily with the temperature maintained between 26.0 °C and 29.0 °C, the pH between 7.0 and 8.0, and the dissolved O$_2$ levels between 5.0 and 10.0 ppm.

### Drug administration

Modafinil (Toronto Research Chemicals) was administered to the fish at three different doses; 2 ($n = 19$), 20 ($n = 24$), or 200 ($n = 22$) mg/L. Modafinil was first dissolved in dimethyl sulfoxide (DMSO; Sigma Aldrich, St. Louis, MO) to increase solubility and this stock solution was frozen until the experimental day. This stock solution was used within 14 days of preparation and mixed with 250 mL of habitat water to achieve the desired dose of modafinil (the maximum amount of DMSO was 0.1%). The control group ($n = 35$) underwent otherwise identical experimental procedures (including 0.1% DMSO) without the presence of modafinil. DMSO does not alter locomotion or anxiety-like behaviour in zebrafish (0.05%: *Sackerman et al., 2010*; 0.1%: *Stewart & Kalueff, 2014*; 0.1%: *Kolesnikova et al., 2017*) so a separate control group was not included in this part of the study. Zebrafish were individually administered modafinil or control solutions for 30 min in 600 ml containers before testing began. Each drug group was performed on two days of testing, with controls interspersed throughout the day at a 2:1 (drug:control) ratio, resulting in ∼$n = 20$ for each drug group and $n = 35$ for the controls. The drug solution was made fresh each day and maintained at 26–28 °C. All of the fish used in this experiment were

A 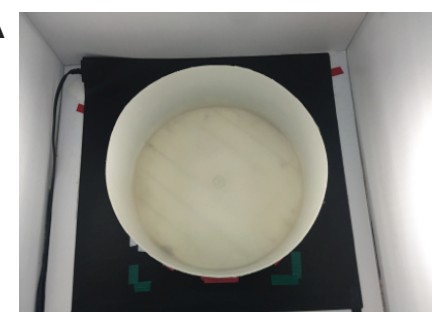 B 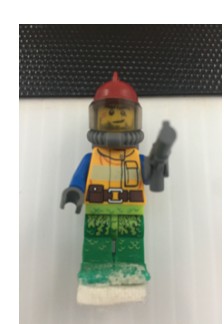

**Figure 1** **Experimental set up.** Apparatus used in the novel approach test. (A) The arena used was a plastic circular arena (diameter: 34 cm, wall height: 16 cm) placed on a seeding mat to maintain water temperature. (B) The novel object was a multicolour Lego figurine (height: 5 cm, width: 1.5 cm).

experimentally and drug naïve. Doses were based on previous studies using modafinil (*Sigurgeirsson et al., 2011*) and other similar psychostimulants (*Sackerman et al., 2010*) in adult zebrafish.

Ethanol is commonly used as an anxiolytic drug in zebrafish studies (*Holcombe, Schalomon & Hamilton, 2014*; *Tran, Facciol & Gerlai, 2016*; *Tran et al., 2016b*) so we used this drug to validate the novel approach test as a measure of anxiety in a second experiment. Ethanol (non-denatured) was added to habitat water in the same 600 ml containers used for modafinil to reach a final concentration of 1% (volume/volume percentage). Zebrafish were individually placed in the solutions for 30 min prior to testing. These experiments were performed on two successive days of testing, with controls interspersed throughout the day at a 1:1 (ethanol:control) ratio, resulting in $n = 20$ for the ethanol group and $n = 19$ for the controls.

## Behavioural testing

Testing was performed during the light hours of 12-hour light/dark cycle. On the day of testing, the fish were transported in their housing tank to the experimental area. The fish were individually placed in either the control or modafinil solution (2, 20, or 200 mg/L) for 30 min. Following drug administration the fish were placed in the circular arena placed into the novel approach test arena; a circular white plastic arena with at diameter of 34 cm and walls that were 16 cm high (Fig. 1A). The arena was filled with 6 cm of habitat water that was maintained at 26–28 °C using a heat mat (Seedling Heat Map—HydroFarm) placed underneath the arena. Water was changed after every 4 fish. A Lego figurine was placed in the middle of the arena during testing and was used as the novel object in this paradigm (Fig. 1B) similar to other studies (*Ou et al., 2015*). It was held in place with a 1.5 cm ×1.5 cm square of Velcro glued to the feet of the Lego figurine and the arena floor. The height of the water completely submerged the figurine. The fish were placed into the arena and recorded for 10 min. Dependent variables were measured using EthoVision XT (version 10; Noldus) motion tracking software. In EthoVision, the arena was divided into three zones, the center, transition, and thigmotaxis zone. The center zone consisted of a 10 cm diameter circle with the middle on top of the object. The thigmotaxis zone was a circular zone from the wall 4.5 cm (one body length of a zebrafish) toward the center of

the arena. The transition zone was the zone in between the center zone and thigmotaxis zone. The variables measured were time in each zone (center, transition, thigmotaxis), distance moved, meandering (the change in direction of movement relative to the distance moved), and duration immobile. To calculate meandering we filtered the trials with the 'minimal distance method' in EthoVision. This eliminates the inclusion of directionality when the fish is still or only minimally moving. Following the behavioural testing the fish were placed back into their housing tanks.

## Statistical analysis

A D'Agostino & Pearson omnibus normality test was used to assess normality for all data sets. One-way ANOVA with a post hoc Dunn's Multiple Comparison Test or unpaired $t$-test was used with parametric data and nonparametric data was analyzed with a Kruskal–Wallis or Mann–Whitney test. Absolute time in each zone was measured as duration (s) in zones for the full 10 min trial. Meandering was quantified using the 'absolute meandering' option in EthoVision. This calculates that total change in angle that the fish moves over the trial in degrees per centimeter. Immobility was defined in as the percent change in the pixels of the fish from frame to frame and was set at a threshold of 5% (*Pham et al., 2009*). The behavioural data was analyzed using GraphPad Prism software (version 6). Data are shown as mean ± s.e.m.

## RESULTS

Zebrafish were exposed to doses of 0 (control), 2, 20, and 200 mg/L for 30 min then immediately tested in the novel approach test. Figure 2 is a representative example of the movement of one zebrafish over the duration of the 10 min trial under control conditions (Fig. 2A) and another zebrafish after being exposed to 200 mg/L of modafinil (Fig. 2B). The heatmaps are examples of the same responses but with pseudocolour representation of the location of the fish over time (Figs. 2C–2D).

### Modafinil: time in zones

To examine the effects of modafinil on anxiety and exploratory behaviour using the novel approach test, the time spent in the center, transition, and thigmotaxis zones were measured. We found that time spent in the thigmotaxis zone was significantly decreased for all modafinil groups compared to control (Fig. 3A; control: 494.2 ± 9.8 s, 2 mg/L: 449.3 ± 15.8 s, 20 mg/L: 441.4 ± 12.5 s, 200 mg/L: 448.1 ± 16.5 s; $F(3, 96) = 4.168$, $P = 0.008$). Time spent in the transition zone was significantly greater for all groups compared to controls (Fig. 3B, control: 101.5 ± 9.3 s, 2 mg/L: 144.9 ± 14.7 s, 20 mg/L: 149.9 ± 11.9 s, 200 mg/L: 144.4 ± 15.1s; $F(3, 96) = 4.136$, $P = 0.008$). There was no significant difference in time spent in the center zone across all groups (Fig. 3C; control: 4.3 ± 0.9s, 2 mg/L: 5.8 ± 1.5 s, 20 mg/L: 8.7 ± 2.5 s, 200 mg/L: 7.5 ± 1.7s; $H(3) = 5.727$, $P = 0.125$).

### Modafinil: locomotion

To examine the locomotor activity of fish dosed with modafinil we quantified the distance moved, meandering, and duration the fish was immobile. The drug treatment groups

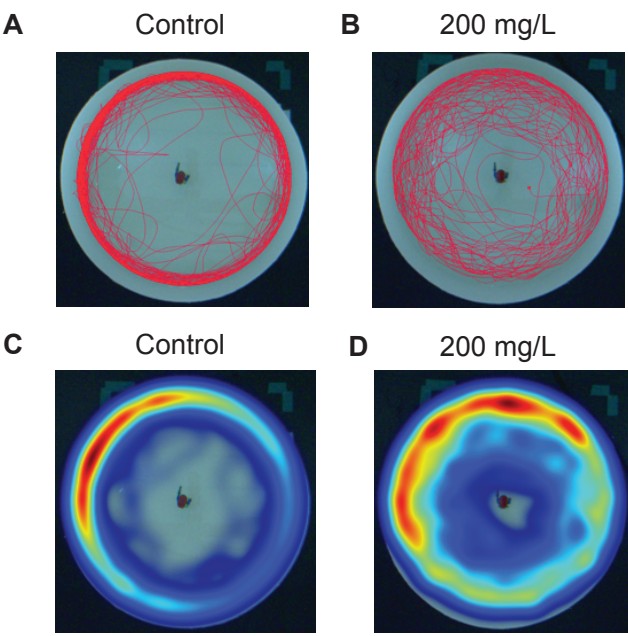

**Figure 2** **Zebrafish movement in the novel approach test.** Individual fish movement in the novel approach test. (A) Trackplot of an individual fish in the control condition for the 10 min trial. (B) Trackplot of an individual fish in the 200 mg/L modafinil condition. (C) Heatmap of the same fish from the control condition in (A). A heatmap is a visualization of the location of the fish over the entire 10 min trial produced in EthoVision. The colours represent the duration of time the fish spent in each pixel with low wavelengths (ex. Red) indicating high time spent and high wavelengths (ex. Blue) indicating low time spent in each pixel. (D) Heatmap for the same fish from the modafinil 200 mg/L condition in (B).

did not significantly differ from the control group in the distance moved (Fig. 4A; control 5,537 ± 164 cm, 2 mg/L: 5957 ± 328 cm, 20 mg/L: 5,410 ± 266 cm, 200 mg/L: 5,288 ± 209 cm, $F(3, 96) = 1.277$, $p = 0.2868$). No significant difference was found for meandering (Fig. 4B; control: 7.9 ± 1.6°/cm, 2 mg/L: 7.9 ± 1.5°/cm, 20 mg/L: 6.3 ± 0.7°/cm, 200 mg/L: 6.9 ± 1.2°/cm, H (3) = 0.6190, $P = 0.8921$) across all groups. There was no significant difference in immobility (Fig. 4C: control: 4.9 ± 1.6 s, 2 mg/L: 3.8 ± 2.2 s, 20 mg/L: 11.8 ± 5.2 s, 200 mg/L: 17.3 ± 7.9 s, $H(3) = 4.032$, $P = 0.2580$) across all groups.

## Ethanol: time in zones

In a separate set of experiments we tested whether ethanol (1%) would also alter the location preference of zebrafish. The time spent in the center, transition, and thigmotaxis zones were measured. We found that time spent in the thigmotaxis zone was significantly decreased for the ethanol exposed fish compared to controls (Table 1). We also found that the time in the transition zone was increased for the ethanol dosed fish compared to controls (Table 1). There was no significant difference in time spent in the center zone between ethanol exposed fish and controls (Table 1).

## Ethanol: locomotion

Ethanol did not have a significant affect on distance moved or meandering (Table 1). However, ethanol did significantly increase the time the fish were immobile (Table 1).

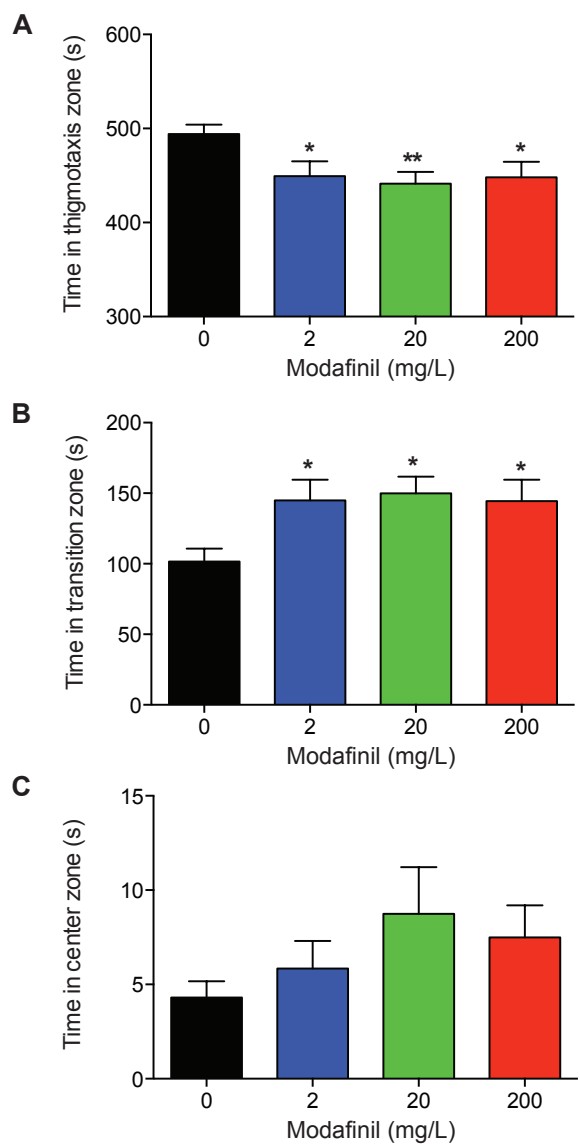

**Figure 3** **Zone preference after modafinil dosing.** Time in zones of the novel approach test. (A) The time spent in the thigmotaxis zone (near walls) of the arena decreased with modafinil. (B) The time spent in the transition zone (middle zone) of the arena increased with modafinil. (C) The time spent in the center zone (where the object was located) was not significantly different across groups. $*p < .05$, $**p < 0.01$, difference from control group.

## DISCUSSION

Our findings show that the acute administration of modafinil decreases anxiety-like behaviour in zebrafish across a range of doses. In particular, zebrafish administered 2, 20, and 200 mg/L modafinil spent significantly less time in the thigmotaxis zone near the walls of the arena, and more time in the transition zone of the novel approach test (Figs. 2 and 3). Ethanol (1%) also increased time spent in the transition zone and decreased time spent in the thigmotaxis zone. This type of behaviour is consistent with decreased

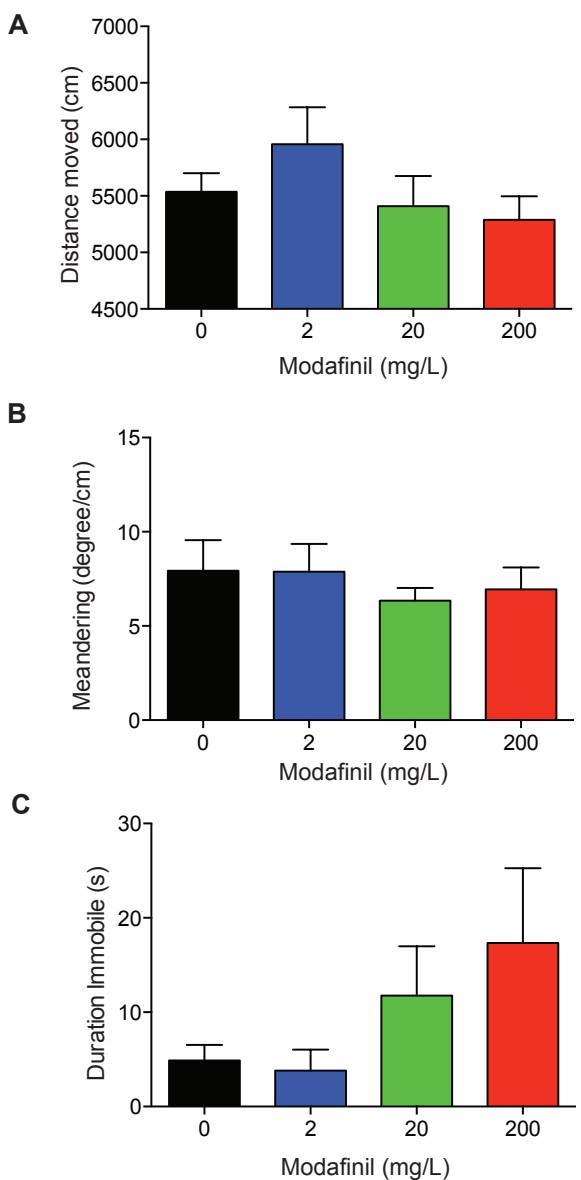

**Figure 4 Locomotion after modafinil dosing.** Locomotion variables in the novel approach test. (A) There was no significant difference in distance moved across all groups. (B) There was no significant difference in meandering across all groups. (C) There was no significant difference in immobility across all groups.

anxiety, and has been observed in zebrafish with anxiolytic compounds resulting in less time near the walls of an open field test (*Maximino et al., 2010*). We observed no significant modafinil-induced changes in locomotion; distance moved, meandering, and immobility, therefore, their zone preference was not due to a locomotor alteration. These results suggest that modafinil decreases anxiety in zebrafish.

The novel approach test involves placing a fish into a circular arena containing a novel object in the center. The innate fear response for a never-before seen object (neophobia) is the behaviour of interest in zebrafish, as they have a tendency to remain away from

**Table 1  Summary of results from ethanol (1%) exposed and control fish in the novel approach test.** Bold items show significant differences between ethanol and control groups ($\alpha = 0.05$). P values were calculated with an unpaired $t$-test for parametric data and a Mann–Whitney test for non-parametric data.

|  | Time in center (s) | Time in transition (s) | Time in thigmotaxis (s) | Distance moved (cm) | Meandering (degree/cm) | Immobility (s) |
|---|---|---|---|---|---|---|
| Control ($n = 19$) | $2.1 \pm 0.5$ | $102.2 \pm 14.9$ | $478.6 \pm 20.3$ | $3,649 \pm 388$ | $5.9 \pm 0.7$ | $58.6 \pm 21.7$ |
| Ethanol ($n = 20$) | $1.9 \pm 0.7$ | $199.5 \pm 35.4$ | $387.4 \pm 35.6$ | $3,216 \pm 357$ | $6.2 \pm 0.3$ | $126.7 \pm 21.8$ |
| Control vs. Ethanol | $P = 0.4167$ | $\boldsymbol{P = 0.0176}$ | $\boldsymbol{P = 0.0345}$ | $P = 0.4167$ | $P = 0.1257$ | $\boldsymbol{P = 0.0090}$ |

the object that they may perceive as a predator (*Wright et al., 2006*; *Moretz, Martins & Robinson, 2007*). This has been pharmacologically validated in juvenile pink salmon with the GABA$_A$ receptor antagonist, gabazine, an anxiogenic compound that decreased time near the novel object and increased time in the thigmotaxis zone (*Ou et al., 2015*). In this study we also validated the novel approach test as a measure of anxiety with the use of ethanol at a dose that has reliably decreased anxiety in other zebrafish studies (*Tran, Facciol & Gerlai, 2016*; *Tran et al., 2016*). We found that fish dosed with ethanol spent less time in the thigmotaxis zone, opposite of the anxiogenic gabazine (*Ou et al., 2015*). Interestingly, we found no difference in the time spent in the center zone with ethanol, suggesting that the exploration or boldness of the fish was not directly increased. Furthermore, in the present study all doses of modafinil decreased time in the thigmotaxis zone, suggesting that anxiety is reduced across all doses. There was a trend towards a decrease in distance moved suggesting a potential dose-dependent effect of modafinil on activity; however, this did not reach significance. These findings are inconsistent with a modafinil-induced increase in rearing and holepoking in mice, which can be considered 'exploration,' and somewhat in contrast with the elevated hyperactivity observed in mice (*Young, Kooistra & Geyer, 2011*).

Modafinil has been shown to occupy bindings sites on the dopamine transporter (DAT) and norepinephrine transporter (NAT) with modest affinity compared to other psychostimulants like methylphenidate (Ritalin; *Madras et al., 2006*; *Minzenberg & Carter, 2008*). This action on the catecholamine system has been shown to increase extracellular dopamine in the medial prefrontal cortex of rats (*De Saint Hilaire et al., 2001*), striatum of mice (*Wisor et al., 2001*) and nucleus accumbens of rats (*Ferraro et al., 1997*; *Murillo-Rodriguez et al., 2007*). In a recent study on the effects of modafinil on dopamine receptor knockout mice the exploratory effect of modafinil was attenuated in *Drd1* and *Drd4* knockout mice, supporting the role of the dopamine D1 and D4 receptors in the increased exploratory response to elevated synaptic dopamine caused by modafinil (*Young, Kooistra & Geyer, 2011*). Extracellular norepinephrine has also been shown to be increased in the prefrontal cortex and hypothalamus and extracellular serotonin increased in the prefrontal cortex and hypothalamus (*De Saint Hilaire et al., 2001*) frontal cortex and amygdala (*Ferraro et al., 2000*; *Ferraro et al., 2002*) of rats. In the current study, we did not perform intracranial microdialysis or use receptor knockouts; however, it is likely that elevated extracellular dopamine, norepinephrine, and serotonin are also at least in part responsible for the anxiolytic effects seen in zebrafish. In other zebrafish research, there is ample evidence that modulation of monoamine neurotransmission leads to changes

in exploratory behaviour and anxiety. Using the norepinephrine reuptake inhibitor, desipramine (25 mg/L), and the serotonin reuptake inhibitor, citalopram (100 mg/L), *Sackerman et al. (2010)* demonstrated that each of these compounds increased time spent in the upper region of a novel tank diving test which is indicative of decreased anxiety. Also using the novel tank diving test, Bencan and colleagues (*2009*) found that buspirone (6.25–50 mg/L) which is mainly a serotonin 1A agonist (with low affinity to the dopamine D2 receptor agonist where it acts as an antagonist, and weak affinity to serotonin 2 receptor) (*Loane & Politis, 2012*) also had an anxiolytic effect. Chronic administration of fluoxetine (100 μg/L for two weeks), a selective serotonin reuptake inhibitor also produced a significant increase in time spent in the top of a novel tank diving test, decreasing anxiety. Future research with specific drugs that decrease dopamine signalling, or genetic knockouts in zebrafish, like the *per1b* which expresses low levels of dopamine (*Huang et al., 2015*) would be beneficial to further understand the mechanism of action of modafinil.

The novel approach test has been considered an appropriate test to assess anxiety in zebrafish (*Wright et al., 2006*; *Stewart et al., 2012*), although, to the best of our knowledge, has never been used with pharmacological compounds in zebrafish (*Maximino et al., 2010*). This research supports the novel approach test as a legitimate test for anxiety and exploratory behaviour in zebrafish and could be used with a variety of compounds. The present findings suggest that modafinil has anxiolytic properties due to the decrease in time the fish spent in the thigmotaxis zone and the increase in time the fish spent in the transition zone. Future research should test the potential nootropic properties of modafinil on zebrafish using learning paradigms like the novel object recognition test (*May et al., 2016*), episodic-like memory test (*Hamilton et al., 2016*), food-based reinforcement tasks (*Ingraham et al., 2016*), or other appetitive conditioning, associative learning or aversive conditioning learning and memory tests (see *Gerlai (2016)* for review). If successful at enhancing memory formation or storage, modafinil could then be administered with a putative model of Alzheimer's disease, the prion protein knockout model (*Fleisch et al., 2013*), in an attempt to rescue these deficits. Once the complex drug profile of modafinil is understood, clinicians may be able to adequately prescribe it to a wide range of patients.

## ACKNOWLEDGEMENTS

We would like to thank Melike Schalomon, Taylor Pitman, Anne Walley, Erica Loh and Aleah McCory (Animal Care Technician) for their assistance with fish husbandry. We are very grateful to Dr. Gregory de Pascale for his helpful comments on this manuscript and to Melissa Bryant for helping with the conception of this study.

### Funding

This work was supported by the Natural Sciences and Engineering Research Council of Canada (NSERC Discovery grant to TJH) grant number 04843. The funders had no role in study design, data collection and analysis, decision to publish, or preparation of the manuscript.

## Grant Disclosures

The following grant information was disclosed by the authors:
Natural Sciences and Engineering Research Council of Canada (NSERC Discovery grant to TJH): 04843.

## Competing Interests

The authors declare there are no competing interests.

## Author Contributions

- Adrian Johnson conceived and designed the experiments, performed the experiments, analyzed the data, wrote the paper, prepared figures and/or tables, reviewed drafts of the paper.
- Trevor James Hamilton conceived and designed the experiments, analyzed the data, contributed reagents/materials/analysis tools, wrote the paper, prepared figures and/or tables, reviewed drafts of the paper.

## Animal Ethics

The following information was supplied relating to ethical approvals (i.e., approving body and any reference numbers):

The experimental procedures were approved by the MacEwan University Animal Research Ethics Board (AREB; protocol number 05-12-13). These standards are in compliance with the Canadian Council for Animal Care (CCAC).

## Data Availability

The raw data has been supplied as a Data S1.

## Supplemental Information

Supplemental information for this article can be found online at http://dx.doi.org/10.7717/peerj.2994#supplemental-information.

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
