# Peer review of "Modafinil decreases anxiety-like behaviour in zebrafish"

_PeerJ, doi:10.7717/peerj.2994_

## Round 0.1 · original submission · Minor Revisions

Our reviewers are very positive about your paper, but suggest that a few more controls might strengthen it considerably. Besides addressing their comments, please explain more fully how the meandering computations were performed. Since a full turn around itself would be at least the length of the fish (i.e. 4.5 cm), I would have expected the meandering values (as turn/angle/distance travelled) to be maximally bound by 360 degrees/4.5cm , i.e. 80 degrees/cm. Since your results state that at 200 mg/mL dose the fish turned by an average value of 814 degrees (i.e. more than two full turns) by cm traveled, I am absolutely sure that my interpretation of the meandering angle is wrong, but I cannot then decide what the correct interpretation would be.

·

Basic reporting

This study investigates the effect of modafinil on zebrafish, using an apparently common thigmotaxis model to assess their level of anxiety. It finds that modafinil appears to make the fish more willing to swim closer to a novel object (Lego figurine) and therefore apparently less anxious. This finding is in accord with some other human and animal trials of modafinil, although contradicting others. They do not find a clear dose-dependent relationship. There was no clear difference in the fishes' locomotor activity.

The paper is correctly structured and decently-written and I have no major issues here. There are a few very minor grammatical issues and typos:

1. Weirdly constructed sentences/excess commas in a few places, eg lines 67-68, line 222, etc. I'm not an editor and not qualified to point out all of these, but they might want to go over again and look for these.

2. In line 139, the word "nonparametric" is spelled "nonaprametic"

3. In line 155 - This may be a complete misunderstanding on my part, but they say that the modafinil groups spent more time in the thigmotaxis zone, yet their data (and hypothesis) seem to suggest that they spent less time in that zone. Suggest rereading and making sure they're saying what they think they are saying.

4. In line 225, there is a sentence which looks like it might have been cut off.

Experimental design

The authors argue that the anxiogenic vs. anxiolytic effect of modafinil is not clearly established. Their study adds to this debate. Although no one would ever directly extrapolate this to humans, it adds to a useful literature on which cases produce which anxiety-related effect that might as a whole eventually provide some insights into the human case.

Validity of the findings

Overall this study looks plausible and solid. They look like they adjust for multiple comparisons. It looks like their sample size is big enough for what they are trying to detect, given their effect size (which seems extraordinarily high to me as someone who works with humans, but which is apparently consistent with other zebrafish studies). The effect is not dose-dependent, but many of modafinil's effects in humans are not very dose-dependent either, so this is not too implausible. Their findings seem to match common sense and (some of) the other studies done in this area.

In general this looks like a good study and I see no reason not to accept it. Note that I have never experimented on zebrafish or other nonhuman animals and I am not qualified to assess whether the authors used correct zebrafish-related methods.

·

Basic reporting

No Comments

Experimental design

a) The authors should substitute the word "dose" for "concentration" throughout the manuscript;
b) What was the DMSO concentration in the habitat water? Were the animals treated at the same time?
c) In my opinion, this study lacks the control group without DMSO. Did the authors check if there are different behavioral responses between these groups?
d) Was the water in the novel approach test changed between each animal?
e) Lines 155-156: “We found that time spent in the thigmotaxis zone was significantly increased for all modafinil groups compared to control”. Please, revise this sentence according to the results observed;
f) I suggest to remove the results description from the figure legends;
g) Different zebrafish behavioral responses are frequently observed depending on the protocol used. I suggest that the authors also evaluate the effects of modafinil in the light-dark test in order to adequately consider the effect of this drug in zebrafish as anxiolytic.

Validity of the findings

No Comments

Additional comments

The aim of this study was investigate the behavioural effects of different concentrations (2-200 mg/L) of modafinil in the novel approach test. This is of significant value in this field of research, since zebrafish is becoming a popular and useful animal model for behavioral neuroscience studies. The manuscript is well written and may be recommended for publication in this journal. For revision, minor changes are suggested.

Reviewer 3 ·

Basic reporting

The paper is well-written and well presented. The figures look fine and the data is easy to follow.

Experimental design

A control experiment using an established anxiolytic is missing. This has been explained in detail below.

Validity of the findings

The conclusions are not that well supported because the test probably requires further validation. This is described below.

Additional comments

This study by Johnson and Hamilton describes the effect of modafinil treatment of adult zebrafish anxiety-like behaviour. Fish treated with an acute dose of modafinil spent more time in the transition zone of this setup with a corresponding decrease in thigmotaxis.

This paper is nicely presented and is exceptionally well written. However I think there are some control experiments that could be included to strengthen this story. As it stands, this research is somewhat limited in its scope – a single experiment is presented – and the use of a classic anxiolytic such as benzodiazepam would be much more convincing. Furthermore, direct measurements of the effects of modafinil on the brain would be helpful. HPLC could be used to look at concentrations of dopamine and noradrenaline after drug application thus demonstrating that modafinil has a conserved effect across species.

Major comments

The novel object interaction test presented here is relatively underused in zebrafish and I am not convinced that it has been sufficiently well developed in this species. The only reference I can find to this is Ogwang et al., 2008 (in table 1 of the review cited here by Stewart et al., 2012). However, this setup is quite different and does not include the three zones used here. The other reference sited to back this up uses juvenile pink salmon. Other tests of anxiety-like behaviour in zebrafish, such as tank diving, have been better described. Whilst it is likely that anxiety is being measured here, a control experiment using a classic anxiolytic would add a lot of weight to the claim that modafinil is reducing anxiety.

The discussion section could also be strengthened by considering two further points. It is not clear why modafinil has opposing actions depending upon the test used and this could be considered in more detail. Secondly, the behavioural test itself could be further discussed. It is likely that several behaviours are being measured here – exploration, novel object boldness and anxiety-like behaviour. Notably neither treatment group enters the centre compartment for much time – does this mean that novel object boldness or exploration is not much changed?

Minor comments

1) Please indicate the strain of zebrafish used in this study.

2) It is not clear how the doses of modafinal were chosen (were these based upon Sigurgeirsson and colleague’s paper?) and why an acute dose was given rather than a chronic dose.

3) The behaviour described here could be describe as “interaction with a novel object” rather than boldness. Boldness can have anthropomorphic connotations. Furthermore, since the fish do not interact with the novel object the behaviour could be better described as a reduction in thigmotaxis.

4) It seems a little odd to maintain the stock drug solution at 26-28 C throughout the day; this risks that the drugs is degraded over time by the temperature. Was there any noticeable different in drug effect on behaviour over the course of the day?

5) I am not that comfortable with the data presented in figure 2 – only showing one fish could in theory bias the presentation of results (although I am not questioning the author’s integrity here). Is it possible to make a heat map based upon the average time spent by all fish? It is also not clear how this image was produced. A colour scheme for the heat map would be useful.

6) Page 11, line 155. Time in the thigmotaxis zone is decreased for modafinil-treated fish, not increased. This makes more sense since time in the transition zone is increased in this group. Line 158 should not read “was also” (same problem here). The trend mentioned for the centre zone should not be mentioned because the differences between treatment groups are tiny (roughly 3 seconds out of a 10 minute experiment). How relevant could this be biologically?

7) Page 13 line 195. I don’t think you can compare meandering in zebrafish to hole poking or rearing in mouse.

8) Page 13 line 205. Mouse genes should be written with a capital letter and in italics (Drd1 etc)

9) Page 13 line 224. I don’t think the per1b mutant would be the most efficient way to examine a reduction of dopamine signalling. It would be difficult to pull apart other per1b-related phenotypes from the changes in DA signalling. Why not pretreat with a drug that reduce dopamine levels? Furthermore this sentence (finishing on line 225) is unfinished.

---

## Round 0.2 · accepted · Accept

You have adequately addressed the comments and suggestions from our reviewers. I am pleased to accept your paper for publication in PeerJ.

·

Basic reporting

No comment.

Experimental design

No comment.

Validity of the findings

No comment.

Additional comments

The new experiments clearly add to the contribution of this paper, and most of my concerns from my previous review have been addressed.